# Alteration of Neural Network and Hippocampal Slice Activation through Exosomes Derived from 5XFAD Nasal Lavage Fluid

**DOI:** 10.3390/ijms241814064

**Published:** 2023-09-14

**Authors:** Sangseong Kim, Jaekyong Jeon, Dulguun Ganbat, Taewoon Kim, Kyusoon Shin, Sungho Hong, Jongwook Hong

**Affiliations:** 1Department of Biomedical Science and Engineering, Gwangju Institute of Science and Technology, Gwangju 61005, Republic of Korea; 2Department of Pharmacy, Hanyang University, Ansan 15588, Republic of Korea; abcde27230@gmail.com (J.J.); dulguun38584@gmail.com (D.G.); 3Department of Bionanotechnology, Graduate School, Hanyang University, Seoul 04763, Republic of Korea; xodns4922@hanyang.ac.kr (T.K.); 23btlsd32@gmail.com (K.S.); 4Computational Neuroscience Unit, Okinawa Institute of Science and Technology, Okinawa 904-0495, Japan; shhong@oist.jp; 5Department of Medical and Digital Engineering, Graduate School, Hanyang University, Seoul 04763, Republic of Korea; 6Department of Bionanoengineering, Hanyang University, Ansan 15588, Republic of Korea

**Keywords:** exosome, nasal lavage fluid, 5XFAD, high-density multielectrode array, hippocampal slice, neural network

## Abstract

Exosomes, key mediators of intercellular transmission of pathogenic proteins, such as amyloid-beta and tau, significantly influence the progression and exacerbation of Alzheimer’s disease (AD) pathology. Present in a variety of biological fluids, including cerebrospinal fluid, blood, saliva, and nasal lavage fluid (NLF), exosomes underscore their potential as integral mediators of AD pathology. By serving as vehicles for disease-specific molecules, exosomes could unveil valuable insights into disease identification and progression. This study emphasizes the imperative to investigate the impacts of exosomes on neural networks to enhance our comprehension of intracerebral neuronal communication and its implications for neurological disorders like AD. After harvesting exosomes derived from NLF of 5XFAD mice, we utilized a high-density multielectrode array (HD-MEA) system, the novel technology enabling concurrent recordings from thousands of neurons in primary cortical neuron cultures and organotypic hippocampal slices. The ensuing results revealed a surge in neuronal firing rates and disoriented neural connectivity, reflecting the effects provoked by pathological amyloid-beta oligomer treatment. The local field potentials in the exosome-treated hippocampal brain slices also exhibited aberrant rhythmicity, along with an elevated level of current source density. While this research is an initial exploration, it highlights the potential of exosomes in modulating neural networks under AD conditions and endorses the HD-MEA as an efficacious tool for exosome studies.

## 1. Introduction

Exosomes have gained considerable attention in neurobiology, offering intricate insights into the multifaceted interactions that govern neural networks [1,2]. These tiny extracellular vesicles are secreted by various neural cell types—neurons, astrocytes, and microglia—all of which have unique roles to play in the functioning of the nervous system. When neurons release exosomes filled with neurotransmitters and synaptic proteins, these vesicles travel to synapses facilitating the synaptic plasticity crucial for learning, memory, and cognitive functions [3]. Astrocytes also secrete exosomes with a slightly different molecular repertoire. These vesicles often contain growth factors, nutrients, and anti-inflammatory agents. Astrocyte-derived exosomes thus assist in maintaining the structural integrity and functional stability of neural networks [4]. They play roles in repairing damaged neurons, fortifying synaptic connections, and reconfiguring existing neural pathways, thus contributing to the adaptability and resilience of networks. In parallel, microglia, often referred to as the immune cells of the nervous system, produce exosomes loaded with cytokines. Depending on the nature of these cytokines, microglial exosomes can have pro-inflammatory or anti-inflammatory effects, impacting neuroinflammation, neuronal survival, and synaptic integrity [5].

While the beneficial aspects of exosomes are fascinating, their potential role in disease pathology, particularly in Alzheimer’s disease (AD), is equally compelling [6,7]. Exosomes isolated from biological fluids like cerebrospinal fluid (CSF), blood, and saliva have been found to carry disease-specific molecules, such as amyloid-beta (Aβ) and tau proteins. These molecules have been implicated in the spread of AD, potentially serving as markers and mediators of disease progression. According to the clinical and experimental evidence, a connection between CSF and olfactory systems is observed, suggesting a new avenue for the role of exosomes in nasal discharge as diagnostic biomarkers [8,9]. The olfactory system, especially the olfactory epithelium, has been shown to display unique features in AD pathology, such as unusual secretase expression and heightened susceptibility to neuroinflammation [10,11]. Given that olfactory dysfunction often precedes cognitive impairment in AD and overlaps anatomically with early AD-affected regions, the potential for applying nasal lavage fluid (NLF) as a non-invasive diagnostic tool is promising. NLF, generated by mucous membranes in the olfactory mucosa, serves to protect these sensory tissues while also potentially capturing the neuropathology unfolding in the surrounding system [12]. Therefore, exosomes obtained from this fluid could offer a snapshot of the ongoing state of neurodegeneration. Due to the easy collectability, it provides an efficient means for early disease detection and monitoring, complementing the information gathered from other conventional biological fluids, such as CSF and blood. Even though the biomarkers in human NLF have been discovered [13], the volume of collectible NLF from mice is significantly constrained, often amounting to less than a milliliter. This limited quantity renders component-based analyses, such as RNA sequencing, technically unfeasible. As a preliminary investigation of this nature, the current study thus concentrates on exploring the overarching functional impact of NLF from AD animal models on live neurons and brain slices.

A comprehensive investigation into the exosomal effects on neural networks has, however, been a daunting task due to inadequate methods. Conventional techniques of detecting neural activity, such as calcium imaging or patch clamping, offer limited data due to the slow responsiveness of calcium dye and the restricted number of neurons that can be recorded [2]. Utilizing a multielectrode array (MEA) system has overcome these limitations by enabling the simultaneous recording of multiple neuronal signals. Some studies have provided critical insights into neural circuit development, signal transduction, gene regulation, and neural synchronization influenced by exosomes derived from various sources [1,14,15]. However, despite their contributions, MEA experiments face inherent constraints, such as a relatively sparse density of electrodes and a considerable gap between electrodes. In this study, we employed a high-density MEA (HD-MEA) featuring 4096 electrodes (64 × 64) to capture the neural network characteristics influenced by exosomes derived from NLF of 5XFAD AD model animals. The superior spatiotemporal resolution of HD-MEA permitted detailed observations of neural ensembles and circuitry [16]. Further analyses, like functional connectivity maps and current source density (CSD) characterization, coupled with conventional neural activity analysis depicted an integrated picture of neural information processing initiated by exosome treatment.

Though the number of repetitions of the experiment and brain slices analyzed were limited in this preliminary attempt, our findings underscore the potential implications of exosomes in modulating neural networks under neurodegenerative conditions. Thus, the results validate the potential of HD-MEA as an effective neurochip for exosome studies, particularly in understanding their effect on neural networks. By investigating the role of exosomes in AD, we can enhance our understanding of disease mechanisms, potentially leading to the development of diagnostic tools, and explore innovative therapeutic strategies for this formidable neurodegenerative disorder.

## 2. Results

### 2.1. Characterization of Exosomes Isolated from the NLF of 5XFAD Mice Using the Flow Amplification Separation Technology (FAST)

To examine the role of NLF-derived exosomes from 5XFAD animals, FAST was used to collect intact exosomes from limited amounts of nasal fluid with high yield and purity [17,18,19] (Figure 1A). Under consistent temperature control, we determined the size distribution of particles isolated from NLF using nanoparticle tracking analysis (Figure 1B). The overall distribution pattern of 5XFAD mice showed higher peaks in particle concentration than control mice. The concentration of 35–205 nm particles isolated from the nasal fluid of 5XFAD mice was approximately 8.5 × 10^8^ particles/mL, which was approximately 3.0-fold higher than that of the control (Figure 1C). According to transmission electron microscopy images, the majority of the collected exosomes were 30–150 nm in diameter (Figure 1D). Considering the same volume loaded onto the grid, the exosome density was higher in 5XFAD mice than that in control mice, as confirmed by a 3.2-fold increase in CD63 exosome marker expression levels (Figure 1E,F). Overall, the secretion of NLF-derived exosomes was enhanced in the AD animal model, similar to other body fluids [12,20]. After the concentration normalization, the neural network in the HD-MEA was measured for both neuronal cultures and organotypic hippocampal slice cultures (OHSCs) with the harvested intact exosomes (Appendix A).

### 2.2. Altered Neuronal Excitability and Network Connectivity in Primary Cortical Neuron Cultures Stimulated with Aβ42 Oligomers and 5XFAD NLF-Derived Exosomes

During the developmental stages of AD symptoms, neuronal excitability has been frequently documented in vitro with pathologic Aβ42 oligomer treatment, as well as in functional magnetic resonance imaging (fMRI) imaging in patients with mild cognitive impairment (MCI). Using HD-MEA, we recorded primary cortical neurons for two weeks to measure the neuronal excitability induced by Aβ42 oligomers (neuronal culture image on CMOS chip in Appendix A). Additionally, 5XFAD NLF-derived exosomes were examined under the same conditions to compare their neuroexcitability. The basic topological properties of neural cultures were assessed on days in vitro (DIV) 7, 10, and 13 based on the number of spikes (NOS), interspike interval (ISI), and mean firing rate (MFR) for each group. During the culture aging process, NOS and MFR levels tended to increase in all groups (Figure 2A,C,D). Among them, the Aβ42 treatment group displayed the strongest activity. The control group exhibited a slightly higher NOS level and a lower MFR than the 5XFAD group. The ISI, indicative of the time interval between two consecutive spikes, tended to decrease with age, implying increased readiness in response to an input (Figure 2E). The average burst duration in the active electrodes decreased monotonically in the control, which was the reversal pattern in the other groups with the aging of cultures (Figure 2B,F). The mean NOS per burst remained consistent across different groups and DIVs, suggesting stable network mechanisms underlying individual burst events (Figure 2G). However, the burst frequency was increased in Aβ42 and 5XFAD NLF groups in older cultures, indicating a more recurrent generation of neuronal activation under these detrimental conditions (Figure 2H). Considering burst patterns in neuronal networks, the network burst duration in the interquartile range showed a steady increase in all groups (Figure 2I).

### 2.3. Differentiating Features of Neuronal Networks in 5XFAD NLF-Treated Neurons

To investigate network properties, it is necessary to apply reliable network parameters to describe their essential characteristics [21]. Among several parameters, we selected clustering coefficients (CCs), node degrees (NDs), and path lengths (PLs), given their credibility and relevance. First, the CC is primarily a measure of the tendency of neurons in a network to form clusters. Network clustering gradually increased in all groups (Figure 3B), albeit with sparser connections in the connectivity map, particularly from DIV 10 to 13 (Figure 3A, Appendix A). The PLs and NDs are presented as average values and distribution histograms for each node (Figure 3C–F). The PL measures the number of steps involved in transmitting signals from one node to another. In the control culture, the PL values steadily decreased (Figure 3C; control, 5.25, 4.23, and 1.00 each for the median values from DIV 7 to 13), suggesting that neuronal communication became more efficient gradually. Given the increasing CCs, this phenomenon can be attributed to the maintenance of longer distances and local shortcut connections [22]. In contrast, PLs in the Aβ42 and 5XFAD NLF groups retained high means and wide deviations, as reflected in the box plots (Figure 3C; Aβ42, 4.14, 3.12, and 3.32; 5XFAD, 4.51, 3.75, and 4.84 for the median values from DIV 7 to 13). Likewise, the PL distribution of the Aβ42 and 5XFAD NLF groups at DIV13 displayed fatter tails compared to the control group, implying inefficient network organization (Figure 3E and Appendix A; control, 1.30; Aβ42, 3.70; 5XFAD, 5.65 for the PL values at the upper 70% cumulative node counts).

These data suggest that networks of the Aβ42 and 5XFAD groups added or maintained more local clustering connections over time, which increased CCs with insufficient long-distance shortcuts, resulting in degraded network efficiency when compared with that of the control. ND, representing the number of connections per node, provided further insights into gradual network changes compared with the control group. In the control group, the ND gradually declined, suggesting a network-wide pruning of connections. However, the Aβ42 group exhibited a substantial increase in the number of connections from DIV 7 to 10, followed by a slight decrease at DIV 13 (Figure 3A,D), and the 5XFAD group showed a similar tendency to a lesser degree. The ND distribution histogram reflects quantitative changes across the DIVs (Figure 3F and Appendix A). In summary, the control group network evolved by losing connections while achieving higher CCs and lower PLs, developing a more efficient, ‘small world’-like structure [22]. However, in both the Aβ42 and 5XFAD NLF groups, the networks added (from DIV 7 to DIV 10) and lost connections (from DIV 10 to 13), although the latter did not demonstrate a considerable improvement in efficiency, possibly due to a disrupted balance between local connections and long-distance shortcuts. This result demonstrates that 5XFAD NLF could induce pathological changes similar to Aβ42 in network activity patterns.

### 2.4. The Effects of Aβ42 Oligomers and 5XFAD NLF-Derived Exosomes on the LFP Property and Oscillation in OHSCs

In contrast to the formation of intrinsic circuitry in the brain, neuronal culture is limited in its artificial construction. To determine the network properties of neurons in a more natural environment, we tested OHSCs incubated with Aβ42 oligomers and 5XFAD NLF-derived exosomes using HD-MEA (Nissl staining image of the cortical slice in Appendix A). Under 4AP stimulation, LFP signals in slices were recorded after incubation for 12 days with each treatment. Due to the deleterious influence of the degenerative factors, it was inevitable to rescue only single active OHSCs for each group from several slices in preparation. Figure 4A presents typical LFP traces for the dentate gyrus (DG) and cornu ammonis (CA) 1 and 2 overlaid on the MEA recording. As in neuronal cultures, internal connections within slices were represented in OHSCs that overlapped on top of the MFR responses (light-blue lines, Figure 4A and Appendix A). After low-pass filtering (<200 Hz), certain oscillations were evident in the DG, CA1, and CA3 of each OHSC (Figure 4A). Based on the LFP analysis, Aβ42 (509.71 µV for the median) exhibited the highest average amplitude, followed by the 5XFAD NLF (370.36 µV) and control (278.82 µV), although the 5XFAD NLF still exhibited the widest deviation range (Figure 4B; control, 279; Aβ42, 25; 5XFAD, 186 for the number of active sites in the recording arena).

Furthermore, the 5XFAD NLF maintained both the highest average value and a wide deviation in the LFP rate (Figure 4C; control, 60.05 ± 0.01 fp/min; Aβ42, 86.66 ± 7.23 fp/min; 5XFAD, 188.50 ± 73.37 fp/min) and duration (Figure 4D; control, 37.34 ± 1.74 ms; Aβ42, 15.71 ± 0.44 ms; 5XFAD, 61.77 ± 5.99 ms). Despite differences in the LFP characteristics, the energy levels in all slices were unaltered, suggesting that the activity dynamics were consistent across test samples (Figure 4E; control, 0.20 ± 0.01 µV·ms; Aβ42, 0.22 ± 0.02 µV·ms; 5XFAD, 0.18 ± 0.01 µV·ms).

### 2.5. Current Source Density (CSD) Analysis to Localize LFP Distribution in OHSCs

Although LFP analysis can deliver characteristic neural activity in brain slices, the vulnerability of the far-field effect to volume conduction is an intrinsic limitation of LFP analysis in terms of signal transmission in recording slices. To overcome this limitation, a second spatial derivative of LFP, the CSD, can be applied to estimate LFP distribution and propagation with high accuracy [23,24]. Comparing mean CSDs across groups, the 5XFAD NLF group exhibited the highest level, implying significantly larger electrical current sources than sinks on the recording slices (Figure 5A; control, 5.26; Aβ42, 5.76; 5XFAD, 7.52 for the median values). Although the mean value of the Aβ42 group was lower than that of the 5XFAD NLF group, the deviation was wider, suggesting that the data for individual slices were complex and spread between the control and 5XFAD NLF samples. The transformation of LFP data into CSD was generated in a time-series representation, illustrating the dynamic interplay of sinks and sources of neural activity. (Appendix A). Representative photographic images were captured during the spread of activity (Figure 5B,C). Parsing of CSD signals separated the sink and source waveforms, where the level and frequency of ridges and furrows of the wave spread were observed (Appendix A). 5XFAD NLF showed the most volatile dynamics, followed by that of the Aβ42-treated OHSCs.

## 3. Discussion

Exosome research, especially in relation to Alzheimer’s disease (AD), has surged in recent years due to their critical role in cellular mediation. Pathological propagation akin to prion diseases has been found in Aβ42, a factor implicated in AD [25]. Despite this knowledge, the exact transfer mechanism of Aβ42 between cells remains elusive. Exosomes carrying Amyloid precursor protein (APP) and Aβ42 have been implicated in the AD pathogenesis, potentially aiding the dissemination of pathological agents [26,27,28]. In this study, we collected nasal lavage fluid (NLF), as it enables non-invasive and convenient procurement of large volumes of nasal washes in clinical settings. Given the strong correlation between olfactory dysfunction and dementia symptoms, we postulated that NLF-derived exosomes may contain AD-related pathological molecules [29,30,31,32]. We employed microfluidic sorting technology to obtain intact exosomes from clean nasal washes in mice [17,33]. However, due to the minimal amounts of exosomes present in the collected nasal washes, the characterization of internal contents faced technical hurdles. Moreover, proteomic or RNA sequencing analysis requires the collection of substantial volumes of nasal wash, thereby posing challenges when testing novel samples for the first time.

Neuronal hyperactivity is a distinctive early-stage AD feature implicated in epileptic seizures in both humans and animal AD models [34,35,36]. Our electrophysiological results concerning the treatment of neuronal cell cultures with soluble Aβ42 oligomers align with previous findings about neuronal hyperexcitability [37,38,39,40]. Surprisingly, treatment with 5XFAD NLF exosomes also amplified neuronal activity, consistent with a prior report on hippocampal hyperactivity using 5XFAD brain slices [41,42]. Based on accumulated evidence, this can be attributed to the attenuation of inhibitory postsynaptic currents and metabolic dysfunction between neurons and astrocytes in 5XFAD hippocampal neurons. Herein, the connectivity between neuronal networks could be systematically determined owing to the exceptional integrity of the HD-MEA recordings from 4096 electrodes in real time. The gradual increase in clustering coefficient values in aging cultures indicates that network stability can also be developed gradually in neurodegenerative conditions, such as Aβ42 oligomer and 5XFAD NLF exosome treatment. According to the quantitative dimensions of the network, this pathological environment stimulates cluster formation within neuronal cultures relevant to neuronal hyperexcitability. Apart from the superficial observation of the network, the qualitative analysis calculated from the PLs between nodes revealed that disintegration of network efficiency occurs during a long incubation period with 5XFAD NLF exosomes at a rate similar to that with Aβ42 treatment. A longitudinal study of soluble Aβ42 oligomers and 5XFAD NLF exosomes in OHSCs was also conducted to determine their neuromodulatory effects in mouse hippocampal slices. We noted increased LFP responses in Aβ42 oligomers and 5XFAD NLF exosomes in OHSCs, which mirrored the increase in neuronal activity observed in the dissociated cultures. The Aβ42 oligomer-treated neuronal cultures exhibited the strongest topological activation characteristics and connectivity. However, the LFP property was higher in 5XFAD NLF exosomes than in those treated with Aβ42 oligomers, which was also reflected in the CSD analysis. In this case, the intrinsic structure of the hippocampal slices possibly consists of neurons and glial cells interspersed within a specific circuit. Altered current conduction in hippocampal slices may result from the interaction of substances in NLF exosomes with diverse cells in the circuitry.

Utilizing HD-MEA recordings, we demonstrated the functional effect of NLF-derived exosomes from AD model animals. Despite being a preliminary trial to explore the influence of exosomes on the neural network of the AD brain, this study presents a novel possibility of NLF-derived exosomes inducing neuronal circuitry reorganization that would be a crucial sign in the disease progression. Applying the innovative technology used in this study, more conclusive results could be derived to characterize the neural network effect of exosomes derived from other types of neurodegenerative animal models, provided a sufficient amount of exosomes and a valid number of brain slices are available.

## 4. Materials and Methods

### 4.1. Animals

B6SJL-Tg (APPSwFlLon, PSEN1×M146L×L286V) (5XFAD) mice were purchased from The Jackson Laboratory (MMRRC Stock No.: 34840-JAX) (Bar Harbor, ME, USA), and experimental procedures were performed according to protocols approved by the Institutional Animal Care and Use Committee (IACUC) of KPCLab (approval number: P171011) (Matthews Urbana, IL, USA) and ©MEDIFRON DBT Inc. (approval number: Medifron 2017-1) (Seoul, Republic of Korea). C57BL/6 mice were obtained from OrientBio Inc. (Gyeonggi, Republic of Korea), and compliance with relevant ethical regulations and animal procedures was reviewed and approved by Seoul National University Hospital IACUC (approval number: 16-0043-c1a0). For the experiments, three animals from each group were used to obtain OHSCs. Two slices were selected based on the LFP activities for control, Aβ42, and 5XFAD in Figure 4.

### 4.2. Nasal Lavage Fluid Extraction

The procedure of NLF extraction was followed by the previous method [33]. After anesthesia, the left ventricle of the mice was cannulated, and blood was cleared by perfusion with cold PBS. Using scissor dissection, the upper airway, including the palatopharyngeal region, was separated, and the mouse head was separated at the larynx level of the upper airway. Two consecutive volumes of 350 µL of PBS were instilled through the pharyngeal opening into the choana. NLF fluids were centrifuged, and supernatants were stored at −80 °C until assayed. To negate the impact of the varying concentrations and scrutinize the effects of 5XFAD and CTL exosomes at equivalent concentrations, a normalization procedure for each exosome sample was implemented. Specifically, the isolated 5XFAD sample was amalgamated with a buffer to equilibrate the concentration of exosomes originating from CTL.

### 4.3. Nanoparticle Tracking Analysis

NTA was performed using an LM10 (NanoSight, Salisbury, UK) instrument. Exosomes separated from minuscule amounts of nasal lavage fluid were diluted with filtered phosphate-buffered saline (PBS) to examine 20 particles per frame and gently injected into the laser chamber. Each exosome sample was subjected to a red laser (642 nm) three times for 1 min; the detection threshold was set to 5 to allow the detection of nanosized particles. The data were analyzed using NTA software (ver. 3.1; NanoSight). All experiments were conducted at room temperature.

### 4.4. Exosome Purification

Nasal-derived exosome isolation was performed as described previously [17,18,19], with minor modifications. In brief, nasal lavage fluid harvested from the mouse model was filtered through a 0.2 µm syringe filter (Sartorius, Goettingen, Germany) to remove aggregates. The nasal fluid, including extracellular vesicles, was carefully collected and kept on ice before performing exosome separation using FAST. To isolate exosomes, the purification buffer was filtered through a 0.2 µm syringe filter. The flow ratio of sample:buffer: magnification was set at 5:95:75. Exosome-sized particles were separated from the other particles, and all samples were maintained at 4 °C during exosome purification.

### 4.5. Western Blotting

CD63 expression was quantitatively analyzed using Western blotting. Purified exosomes were mixed with 5% sodium dodecyl sulfate (SDS) sample buffer (Tech & Innovation, Gangwon, Republic of Korea), and sliced tissues were lysed and homogenized in radio-immunoprecipitation assay buffer (Tech & Innovation, Republic of Korea) containing protease inhibitors. The protein concentrations of the separated solutions were measured using the Bradford assay (Bio-Rad, Hercules, CA, USA). The samples were heated for 10 min at 97 °C. Next, 15 µg of protein from each sample was subjected to SDS-polyacrylamide gel electrophoresis (PAGE) (12%) and transferred to polyvinylidene difluoride (PVDF) membranes (Bio-Rad) for 90 min. Each membrane was blocked with 5% skim milk (BD Life Sciences, Franklin Lakes, NJ, USA) in TBST buffer (25 mM Tris, 190 mM NaCl, and 0.05% Tween 20, pH 7.5) for 1 h at room temperature, followed by incubation with primary antibodies (anti-CD63, 1:300; Novus Biologicals, Centennial, CO, USA) at 4 °C overnight. After five 20 min washes with TBST, each membrane was washed three times in Tween-20 and incubated with goat anti-mouse IgG for 2 h. Bands were visualized using an enhanced chemiluminescence system; the intensity of the blots was quantified with Image J 1.44 software (https://imagej.nih.gov/ij/plugins/index.html) [43].

### 4.6. Transmission Electron Microscopy

Separated exosomes were diluted and fixed with 2% glutaraldehyde overnight at 4 °C. The mixture of exosomes and fixation solution was then diluted 10-fold with PBS for electron microscopic observation. Briefly, 5 µL of each sample was plated onto a glow-discharged carbon-coated grid (Harrick Plasma, Ithaca, NY, USA), which was immediately negatively stained using 1% uranyl acetate. The exosome samples on the grids were observed under a Tecnai 10 transmission electron microscope (FEI, Hillsboro, OR, USA) operated at 100 kV. Images were acquired with a 2K × 2K UltraScan CCD camera (Gatan, Pleasanton, CA, USA).

### 4.7. Aβ42 Oligomer Preparation

The peptide corresponding to human Aβ42 (Anaspec, Fremont, CA, USA, AS-64129-1, 1 mg) was dissolved in 100 μL of DMSO by vortexing for 30 min at room temperature, and then the solution was added to 900 μL of PBS for incubation at 4 °C for 24 h.

### 4.8. Primary Neuron Culture

Dissection medium Neurobasal Media (NBM) consisted of 45 mL Neurobasal Medium A, 1 mL B27 (50 X), 0.5 mM Glutamine sol, 25 μM Glutamate, 5 mL Horse serum, and 500 μL penicillin/streptomycin. And the culture medium consisted of 50 mL Neurobasal Medium A, 1 mL B27, 0.5 mM Glutamine sol, 500 μL penicillin/streptomycin, and 50 μL HEPES. The Biochip chamber (3Brain, Arena, Houston, TX, USA) was cleaned, filled with 70% ethanol for 20–30 min, rinsed with autoclaved DDW 3–4 times, and, dried in the clean bench overnight with NBM. On the day after, 30–90 μL filtered PDLO which dissolved in borate buffer on the active surface of the Biochip was added and placed overnight in the incubator. The Biochip was washed with autoclaved DDW 3 times before cell seeding. Primary cortical and hippocampal neuron cultures were prepared from postnatal 0-day mouse pups. The pups were decapitated with sterilized scissors and the whole brain was removed. The removed brain was chilled in a cold neurobasal medium with papain 0.003 g/mL solution at 4 °C in a 35 mm diameter dish. Surrounding meninges and excess white matter were pulled out under the microscope (Inverted microscope, Nikon, Tokyo, Japan) in the same solution and transferred to the second dish at 4 °C. The cortex and hippocampus parts were isolated from other parts of the brain, washed with NBM and papain solution, and minced into small pieces. The minced tissues were transferred into a 15 mL tube and incubated for 30 min in the 37 °C water bath. After, the tube was inverted gently every 5 min to be mixed. The tissues were washed with HBSS twice, after being settled down, and the cortex and hippocampi tissues were transferred into prewarmed NBM and triturated 20–30 times using a fire-polished Pasteur pipette. The number of cells was counted and 30–90 µL drops of the cells were plated in the Biochip, which contains ~1000–1500 cells/µL (incubated at 37 °C in 5% CO_2_). From day 2 of the culture, the whole medium was replaced with a fresh feeding medium every 3 days. A quantity of 10 µM of Aβ42 oligomers was treated at DIV 7, 10, and 13. 5XFAD NLF exosomes were incubated from DIV 7 until DIV 13.

### 4.9. Organotypic Hippocampal Slice Culture

The dissection medium consisted of 40 mL Hibernate A, 0.5 mM L-Glutamine, and 10 mL Horse serum. Growth medium 1 consisted of 40 mL Neurobasal A, 20% Horse serum, and 400 μL penicillin/streptomycin solution. Growth medium 2 consisted of 40 mL Neurobasal A, 2% B27, and supplement, 400 μL penicillin/streptomycin solution. The organotypic hippocampal slice culture was prepared from postnatal 7-day mouse pups. Decapitation was performed under cervical dislocation, and the entire brain was removed with forceps and chilled in Hibernate A medium for 10 s at 4 °C. After that, the excess white matter and meninges surrounding the brain were removed under the microscope (inverted microscope; Nikon, Japan) carefully and the hippocampus was dissected with a spatula. The dissected hippocampus was placed on the tissue chopper instrument (Stoelting tissue slicer) which has filter-paper coverage and sliced transversally to 300 µm. The freshly cut sections were collected into the cold dissection medium and separated from each other with a spatula. From the best slices, up to 4 slices were transferred onto the cold culture membrane (Millicell membrane, 0.4 μm), which was then placed into the prewarmed 6-well plate with growth medium 1. From day 2, the medium was changed with growth medium 2, and the whole medium was replaced every 3 days. Quantities of 10 µM of Aβ42 oligomers and 5XFAD NLF exosomes were treated at DIV 7 until DIV 13.

### 4.10. Neuronal Spike and LFP Recording with the High-Density Multielectrode Array (HD-MEA)

High-density multielectrode array (HD MEA) recording with 4096 electrodes in a CMOS Biochip (BiocamX, 3Brain GmbH, Pfäffikon, Switzerland) was conducted at a sampling rate of 9 KHz. The active electrode, which is 21 μm × 21 μm in size and 42 μm in pitch, is implanted in the array with a 64 × 64 grid (2.67 × 2.67 mm^2^) centered in a working area (6 × 6 mm^2^). The brain slice was positioned at the center of the Biochip under the square-shape platinum net anchor to prevent it from displacement by the perfusion flux. To overlay the real image of the slice at the site of recording, a stereomicroscope with transmitted illumination was settled over the slice with 20× magnification (Nikon, SMZ745T, Japan). The neuronal activities in cortex slices were recorded for approximately 35 min with and without additional chemicals. To generate spontaneous epileptic-like discharges, the slices were perfused with Kv1 channel blocker 4-Aminopyridine (4AP) 250 μM (Sigma-Aldrich, Saint Louis, MO, USA). The spontaneous response was recorded for the first 5 min following the oxygenated 4AP for up to 15 min. All recordings were conducted by Brainwave 5 software (3Brain GmbH, Switzerland; https://www.3brain.com/products/software/brainwave5).

### 4.11. HD MEA Data Analysis

Raw data were filtered with an IIR low-pass filter (cutoff at 200 Hz, order 5) before LFP detection. To identify LFP events, a standard hard double-threshold algorithm was used (upper threshold, 40 µV; lower threshold, −40 µV). When the signal overcame one of the two thresholds, an LFP was detected. To determine the duration of the LFP, the energy of the signal was calculated on a sliding window of 50 ms moving forward and backward around the peak until the energy was 1.5 times higher than the energy calculated on the noise. A refractory period (i.e., the minimum distance between two consecutive LFPs on an electrode) of 50 ms was set by the operator. An electrode was considered active if the LFP rate was at least 0.05 event/s. After detection, statistics on the LFP features were extracted by averaging the parameter on each electrode along the recording. The distribution of the feature was calculated by grouping electrodes belonging to the same anatomical area. To identify the different areas concerning the electrode’s position, an image of the recorded slice was superimposed on the map of the electrode grid, allowing manual selection of electrodes for each area of interest. LFP events generally involved most of the area of interest, so to get rid of spurious false-positive detected events, an automated cleaning procedure was used before feature extraction. Detected LFPs were considered valid only if they occurred simultaneously on at least 40% of the total electrodes of an area within a time window of 300 ms. All the parameters from spike detection and LFP recording were calculated with brainwave 5 software (3Brain GmbH, Switzerland; https://www.3brain.com/products/software/brainwave5) [44].

### 4.12. Current Source Density (CSD) Analysis

In the two-dimensional CSD analysis, we first preprocessed the data in two steps: first, at each time point, we identified the saturated electrodes by simple thresholding and inpainted missing signals by linear interpolation with the data from surrounding regions. We monitored how much space the saturated electrodes occupied and checked whether this procedure caused any noticeable artifacts. Then, we smoothed the data spatially by a Gaussian kernel with σ = dpitch. From this, the current source was estimated by applying the modified two-dimensional Laplacian:$$\begin{array}{ll} CSD(x,y,t) & =-\nabla^{2}\phi (x,y,t)\\ & \approx -\frac{2}{3}\cdot \left[\phi (x\pm \Delta x,y,t)+\phi (x,y\pm \Delta y,t)-4\phi (x,y,t)\right]\\ & \;-\frac{1}{6}\cdot \left[\phi \left(x\pm \Delta x,y\pm \Delta y,t\right)+\phi \left(x\pm \Delta x,y\mp \Delta y,t\right)-4\phi \left(x,y,t\right)\right]. \end{array}\;\;$$

From this, we isolated the CSD within the region of interest (ROI) and time window of interest (TOI), computed in the following way: We first estimated the dynamic amplitudes of LFP signals, *A*(*x*,*y*,*t*), by applying Hilbert transformation (MATLAB function *hilbert*) and computing their absolute values. Then, for each (*x*, *y*), we computed the amplitude variability, *ξ*(*x*,*y*) = STD[*A*(*x*,*y*,*t*)]*_t_*, and the ROI was selected by a criterion, *ξ*(*x*,*y*) > Median[*ξ*(*x*,*y*)] + 2.326 STD[*ξ*(*x*,*y*)]. For each (*x*, *y*) in the ROI, a TOI was selected by *A*(*x*,*y*,*t*) > μ_noise_ + 2.326 √2 σ_noise_, where μ_noise_ = ${\langle \overset{-}{A}\left(x,y\right)\rangle }_{\left(x,y\right)\notin ROI}$, σ_noise_ = STD${\left[\overset{-}{A}\left(x,y\right)\right]}_{\left(x,y\right)\notin ROI}$, and $\overset{-}{A}(x,y)$ is the temporal average of *A*(*x*,*y*,*t*). Then, the average rectified CSD (rCSD) was computed by averaging the absolute value of the CSD, $\left|CSD(x,y,t)\right|$, within the ROI and TOI.

All CSD analysis was performed by custom scripts in MATLAB 2018a (Mathworks Inc., Natick, MA, USA), which will be made available upon request.

## 5. Conclusions

Applying HD-MEA, our study illuminates the impacts of 5XFAD NLF-derived exosomes on neural networks, expanding our comprehension of intracerebral communication and its implications for AD. Analogous to the effects of amyloid-beta oligomer treatment, the exosomes result in the biphasic neural activation reminiscent of MCI patients’ neurophysiology. These findings underscore the implication of exosomes in AD, additionally introducing a novel possibility of HD-MEA as an efficacious method for the study of neural networks.

## Figures and Tables

**Figure 1 ijms-24-14064-f001:**
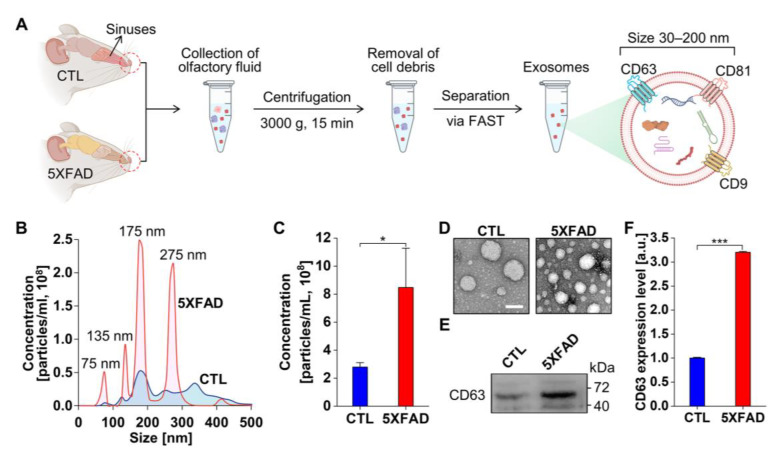
The workflow of exosome preparation from NLF and its quantification following functional tests in HD MEA. (**A**) Flow amplification separation technology (FAST) method to collect intact exosomes from NLF of 5XFAD and control (CTL) animals. Tetraspanins expression in the vesicle membrane, such as CD63, CD81, and CD9, as markers of exosomes. (**B**) Size distribution of particles isolated from NLF through nanoparticle tracking analysis. NLF exosomes from 5XFAD are in red with those from CTL in blue. (**C**) Particle concentration in the range of 35~205 nm. (**D**) Transmission electron microscope (TEM) images in the grids of the same volume loading with NLF-derived exosomes from 5XFAD and CTL. Scale bar = 100 nm. (**E**) Western blot of the exosome surface marker CD63. (**F**) The protein expression level of exosome surface marker CD63. The values are means ± SEMs from three independent experiments. *** *p* < 0.005, * *p* < 0.05; unpaired, two-tailed *t*-test with Welch’s correction.

**Figure 2 ijms-24-14064-f002:**
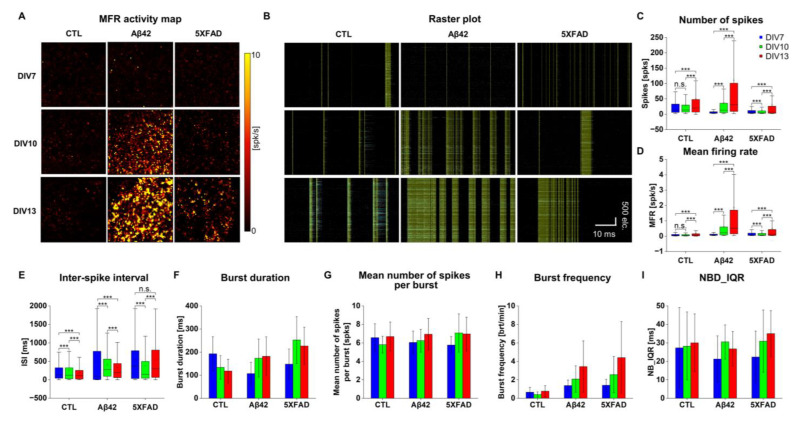
Topological properties of neuronal cultures on HD MEA recording with culture development. (**A**) MFR activity map in spontaneous neuronal activation during 60 s recordings represented from the 64 × 64 grid electrode arena (2.67 × 2.67 mm^2^). Control, Aβ42, and 5XFAD NLF treatment groups on DIV 7, 10, and 13. The intensity scale ranges from 0 to 10 spikes per second. (**B**) Raster plot of neuronal spiking from 4096 electrodes (y-axis) for 60 s recording (x-axis). Mustard color represents spike network bursts and blue represents spike bursts. (**C**) Box plot of the number of total spikes for 60 s recordings in the individual culture chips. Blue for DIV 7, green for DIV 10, and red for DIV 13. (**D**) Box plot of mean firing rate (MFR) in spikes per second. (**E**) Box plot of interspike interval (ISI) in milliseconds. (**F**) Average burst duration with error bars in milliseconds. (**G**) Average of the mean number of spikes per burst in the raster plot with error bars. (**H**) Average of burst frequency in bursts per minute with error bars. (**I**) Average of network burst in interquartile range in milliseconds. In all the box plots above, the lower quartile as the borderline of the box nearest to zero expresses the 25th percentile, whereas the upper quartile as the borderline of the box farthest from zero indicates the 75th percentile. Error bars show SEMs. *** *p* < 0.005; unpaired, two-tailed *t*-test with Welch’s correction; n.s. not significant.

**Figure 3 ijms-24-14064-f003:**
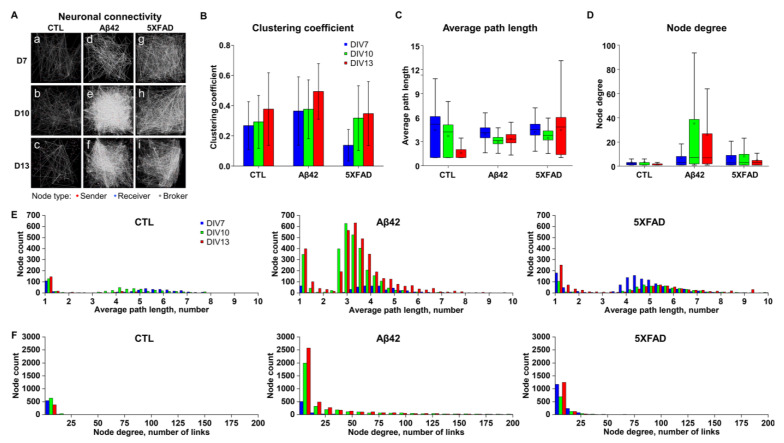
Characteristics of neuronal connectivity in networks. (**A**) Neuronal connectivity map in spontaneous neuronal activation during 60 s recordings represented from the 64 × 64 grid electrode arena (2.67 × 2.67 mm^2^). Control (CTL; a–c), Aβ42 (d–f), and 5XFAD (g–i) NLF treatment groups on DIV 7, 10, and 13. The red dot represents a node of the sender, blue the receiver, and gray the broker. The white lines describe the connections between nodes. (**B**) Average clustering coefficients from neuronal connections for 60 s recordings in the individual culture chips. Blue for DIV 7, green for DIV 10, and red for DIV 13. (**C**) Box plot of average path length in a number of links. (**D**) Box plot of node degrees. (**E**) Distribution histogram of node counts to average path length in number. (**F**) Distribution histogram of node counts to node degree in a number of links.

**Figure 4 ijms-24-14064-f004:**
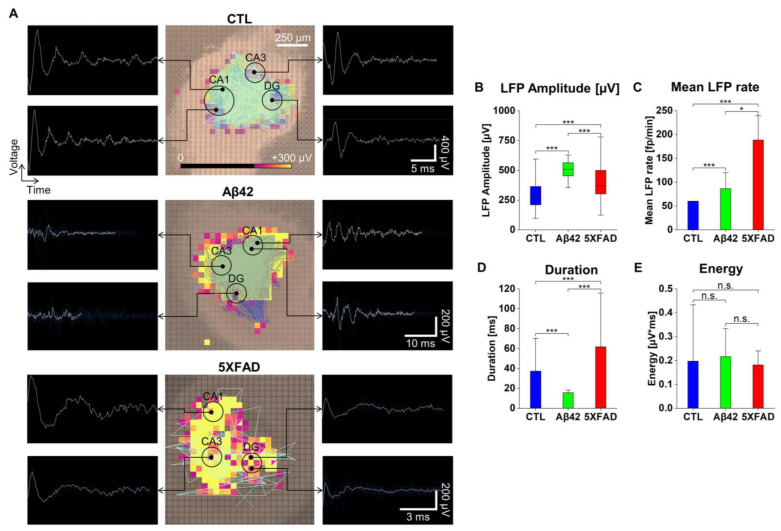
MEA record of LFP responses in organotypic hippocampal slices (OHCSs). (**A**) Overlay of LFP activities over the OHCSs with distinctive waveforms in DG, CA1, and -2 regions represented from the 64 × 64 grid electrode arena (2.67 × 2.67 mm^2^). Control, Aβ42, and 5XFAD NLF treatment groups after incubation of 12 days. The intensity of LFP activity in the color scale from 0 to 300 µV. The white waveforms inside the black columns represent LFP signals from the specific locations denoted by arrows. The sky-blue lines overlaid on the slice images represent connections within the slices. LFP waveforms after the low-pass filter (<200 Hz). (**B**) Box plot of LFP amplitude in micro voltage. (**C**) Average of mean LFP rate in number of LFPs per minute with error bars. (**D**) The average duration of LFPs in milliseconds with error bars. (**E**) Average of mean LFP energy in charge unit of measurement for the area under a voltage–time curve. Error bars show SEMs. *** *p* < 0.005, * *p* < 0.05; unpaired, two-tailed *t*-test with Welch’s correction; n.s. not significant.

**Figure 5 ijms-24-14064-f005:**
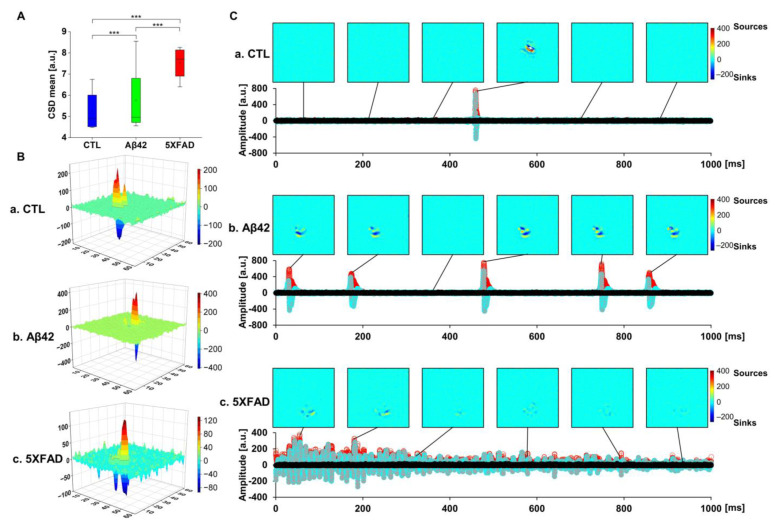
CSD propagation in organotypic hippocampal slices (OHCSs). (**A**) Box plot of CSD means in arbitrary units (a.u.). (**B**) Snapshot of CSD amplitude changes in 3D graph captured from Appendix A of respective CSD propagations. Blue for sinks and red for sources in the scale bar. (**C**) Distribution of CSD incidences over time in amplitude. The upper images were captured from Appendix A of the respective CSD propagations. Blue for sinks and red for sources in the scale bar. *** *p* < 0.005; unpaired, two-tailed *t*-test with Welch’s correction.

## Data Availability

All the data can be shared through the database of MDPI.

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
