# Peer review of "Alteration of Neural Network and Hippocampal Slice Activation through Exosomes Derived from 5XFAD Nasal Lavage Fluid"

_ijms, 2023, doi:10.3390/ijms241814064_

Round 1

Reviewer 1 Report

The manuscript submitted by Sang Seong Kim et al  describes a technically impeccable and well-described work to measure the effect of exosomes from the nasal fluid of mice on the activity and connectivity of cultured neurons and brain slices, but the relevance and main objective of the work as well as the  experimental controls are not entirely clear, so in order to reach publication quality, It would be convenient for the authors to take into account the following issues:

1-The introduction seems superficial; the origin, composition, and function of exoxomes in the nervous system should be expanded with more complete information and concise data.

2-To understand the objective of the work and the choice of nasal fluid exoxomes, is necessary to position these exoxomes in a physiological context, where  are they generated?, where do they go?, are they representative of those of other fluids?...etc.

3_The fact that NLF exosome content cannot be known due to quantity limitations, should be  clrearly stated since the beginning of the work. In no case can conclusions be taken based on an unknown composition, not even in speculation mode, and this should be made clear so as not to lead to confusion.

4_ Drawings of figure 1 A have nothing to do with the work. Lines 79-82 to be erased.

5_ It is not clear how the control neurons are treated, it is inferred that they are NLF exosomes from control mice, but which quantity is used? this is an important point because the controls have 3 times fewer exoxomes and are of different sizes. So it would be understood that cells or brain slices are treated with equivalent amounts because otherwise the results might not be comparable. Other controls are maybe needed and this point needs to  be discussed.

Reviewer 2 Report

ijms-2576747: “Alteration of neural network and hippocampal slice activation through exosomes derived from 5XFAD nasal lavage fluid”

In this study, the authors are trying to demonstrate a potential of nasal lavage fluid analysis in the unveiling of a role of exosomes in mechanisms of neuronal networks function at Alzheimer’s pathology in mice. The material is described successively and conclusions are partially supported by obtained data.

Remarks/recommendations:

1.    “4. Materials and Methods” in the main text should contain a detailed description of the material from “Supplementary Materials and Methods”;

2.    In the “Supplementary Materials and Methods”, a) the number of animals and slices should be clarified, b) “BD” should be open, c) “Image J software” needs a reference, d) what “(ref2)” means, e) (Stoelting tissue slicer 51425, SA), f) “   software (3Brain)…” needs a reference.

3.    In the text, no reference(s) to Figure 1A;

4.    In line 85, “2. Results” should be removed while the same in line 79 should be as a paragraph;

5.    In line 102, “OHSCs” should be open;

6.    In the legend to Figure 1, a) “CTL” should be clarified; b) “…exosomes. (C)…”, c) “TEM” should be open;

7.    In line 108, “MCI” should be open;

8.    In the legend to Figure 2, “ *** “should be clarified;

9.    In Figure 3, a) the colored dots on plate “A” are invisible, b) all plates should be denoted by the letters, c) “CTL” should be clarified;

10. In Figure 4A, the “black” columns should be entitled by “LFP”;

11. In the legend to Figure 4, “  **p < .01  ” should be removed;

12. In Figure 5, all plates should be denoted by the letters;

13. In the legend to Figure 5, “… arbitrary unit (a.u.).“;

14. In line 274, “…at - 80oC…”;

15. In lines 290 and 291, “…mm2…”;

16. In “References”, a) duplicated enumeration, b) no more than five authors should be denoted, c) no titles of the articles;

17. English needs to be double-checked.

Minor editing of English language required

Round 2

Reviewer 1 Report

Authors have improved the most doubtful aspects of the introduction and methodology. The work is ready for publication

Author Response

We appreciate to your effort concerning the revision process.

Reviewer 2 Report

ijms-2576747: “Alteration of neural network and hippocampal slice activation through exosomes derived from 5XFAD nasal lavage fluid”

The authors have made a careful revision and responded to almost all points I raised.

Residual remarks:

1.    In the legend to Figure 6, “ *** “should be clarified (see plate A).

2.    The legends to figures should be presented in a format, which should be similar to that in the body text.

Author Response

  1. In the legend to Figure 6, “ *** “should be clarified (see plate A).

"***p < .005; unpaired, two-tailed t test with Welch's correction."

This sentence was inserted at the end of the legend.

  1. The legends to figures should be presented in a format, which should be similar to that in the body text.

All the legends were presented in the body text format instead of caption.